# Workplace Flexibility for Sustainable Career Satisfaction: Case of Handling in the Aviation Sector in North Cyprus

Huseyin Karsili [1,*], Mehmet Yesiltas [1] and Aysen Berberoglu [2]

1 Department of Business Administration, Faculty of Economics and Administrative Sciences, Cyprus International University, 99258 Lefkoşa, North Cyprus, Turkey; myesiltas@ciu.edu.tr
2 Faculty of Business Administration, University of Mediterranean Karpasia, 99010 Lefkoşa, North Cyprus, Turkey; aysen.berberoglu@akun.edu.tr
* Correspondence: hso_21@hotmail.com; Tel.: +90-5338650242

**Abstract:** The purpose of this research was to find out how workplace flexibility affects the employees' flexibility in order to increase their career satisfaction while reducing their workplace stress with the mediating role of goal orientation. Employees need workplace flexibility to develop a better sustainable career. In doing so, the relationship between workplace flexibility and career satisfaction can be affected by two different factors. One of them is job stress, which can be a mediating factor, and the second is goal orientation, which in this study was considered as a moderator between two variables. For this research, a quantitative research method was applied, and a survey was distributed to 216 respondents, namely, everyone working in handling in a single aviation sector of North Cyprus, to obtain better and clearer results from the respondents. A pilot test was completed and data were collected face-to-face in order to observe the reaction of respondents to develop better results and reduce any mistakes that could arise by answering the questionnaire. Moreover, in order to test the reliability of questionnaires, a pilot test was completed with 14% of the respondents and the results were evaluated by examining Cronbach's alpha. Job stress is a negative term; therefore, surprisingly, there was a positive correlation between workplace flexibility and job stress in the findings. The results were discussed and specifically analyzed with the literature review. Findings of the article clarify that workplace flexibility, along with goal orientation, is expected to positively contribute to the sustainable career satisfaction of employees in the handling sector. This research will make an important contribution to the existing literature pertaining to flexible arrangements in the workplace, sustainable career satisfaction, job stress, and goal orientation, and will contribute to further theories in this field.

**Keywords:** workplace flexibility; sustainability; career satisfaction; job stress; goal orientation

## 1. Introduction

The main purpose of this paper was to clarify the impact of workplace flexibility on the employees' career satisfaction and the contribution of job stress and goal orientation to this relationship. The objective of the study was to reach conclusions which can help the management of the aviation sector, specifically to improve the career satisfaction of the employees working in North Cyprus at its single airport. According to Raziq and Maula-bakhsh [1], workplace flexibility has a beneficial contribution to the employers. Employees require a sustainable environment to reduce their work-stress in order to increase career satisfaction. However, employees have struggled to find useful tools for workplace flexibility provided by their employees. In this field of research, career satisfaction has been debated for numerous years. Park [2] argues that although the career environment is changing, and is mostly focused on employees' liability and the personal behaviors of career growth, subjective sustainable satisfaction of an employee's career within the workplace converts it into a permanent rule in order to have successful career. However, sustainable satisfaction of the career should be discussed based on the employee's career perspective and workplace

actions. Although employees' points of view, actions, and earnings have been debated in the current literature, there are limited variables regarding career satisfaction related to the workplace. Flexibility is known as a major concept of the modern age workplace (Bal et al. [3]). Mejri et al. [4] adds that even though flexibility is clarified enough in the current literature for the process of business, it has become problematic in tangible measurable terms to explain the flexibility of a business process. Additionally, Martínez et al. [5] stated that some of the practices which are an effect of workplace flexibility, such as employees having transient contracts or teams that are multi-purpose, also aim to decrease the costs to the company. It is expected that the desire for workplace flexibility will increase globally while employees are growing older and the working demographic changes. These changes cost administrations, businesses, and workers considerably; to improve the sustainability of employees' careers and workplaces requires careful redesigning and redevelopment in the near future. According to Zeytinoglu [6], from the beginning of the 1990s, there has been extraordinary employment growth, necessitating employers to develop a new category for extraordinary employment opportunities. Moreover, in this situation, the most common apprehension is to develop different modes of employment. These include employment for permanent employees and extraordinary employment. Yazgan, E. [7] states that the main factors of sustainability must be considered and analyzed by employers similar to other problems in business. At a corporate management level, these factors have substantial influence. In order to achieve objectives of sustainability at an institutive level, airline corporations need to have robust sustainability constraints for successful environmental, social and economic outcomes. Additionally, is an extraordinary job actually a bad job? Park [2] also added that the current literature points towards the concept of a sustainable career satisfaction for the next century. Due to this shift, careers are becoming more focused on employees' liability and methods to help manage and control their own behaviors that lead to self-growth. Satisfaction of a sustainable career is explained as employees' work environment and workplace. It is also known as a specific valued career within the workplace. An objective way of defining a successful career is the main focus of researchers while researching sustainable career success. Furthermore, for employees, it becomes more important to set standards for their own careers. Svensson, G., and Wagner, B. [8] expressed sustainability in business as an organization's administrative capabilities, especially their influence on life and the eco-system at a global level within the network of international business practices. Alameeri et al. [9] clarified that the aviation sector has become an important tool for tourism, and has also increased the competitive level of economic and social problems. Most of the information within the current literature correlates sustainability in the aviation sector as an indicator of service, quality, and cost. Alameeri et al. [9] also added that practices which have been implemented in workplaces are directly connected with economic and technological ventures in order to manage top-level sustainable performance and sustainable economic growth for airline corporations. Al Sarrah et al. [10] suggested that the balance between efficiency and sustainability goals should now be considered as the approach to deal with all dimensions of stakeholder engagement. Increasing economic, environmental, and social costs of the civil aviation sector should be the main concern to the sector's management as the sector grows; however, sustainability in the civil aviation sector has not yet been sufficiently addressed. Employees express more satisfaction whenever their employers allow them to pursue and achieve career goals. Usually, in reviewed studies, researchers have considered sustainable career satisfaction as a social link, stressor, and a statement. Contemporary researchers have also argued about how useful behaviors inside the workplace and backing are for sustainable career research.

Therefore, this article identifies the relationship between workplace flexibility and sustainable career satisfaction in the aviation sector. Moreover, the specific objectives of this paper can also be expressed as:

1.  To study the type of flexibility provided in this contemporary era in the airline sector of North Cyprus;

2. To analyze the impact of reduced job stress on employees' sustainable career satisfaction in the aviation sector of North Cyprus;
3. To analyze the impact of workplace flexibility on employees' sustainable career satisfaction in the aviation sector of North Cyprus;
4. To understand the influence of adapting goal orientation perception to sustainable career satisfaction;
5. To recommend the ways through which employee sustainable career satisfaction can be increased with flexible work environments with less stress on employees in the aviation sector of North Cyprus.

The main aim of this article is to understand the relationship between workplace flexibility and career satisfaction, as well as to recognize how this relationship is mediated by job stress and moderated by goal orientation. Considering that it is crucial for organizations to sustain satisfied and motivated employees, it is important for organizations to provide them with career satisfaction. By doing so, organizations should understand what kind of factors affect the career satisfaction of employees. These factors have become increasingly important, particularly in the aviation industry, where customer satisfaction is solely dependent on employee motivation.

## 2. Literature Review

### 2.1. Workplace Flexibility, Career Satisfaction, Job Stress, Goal Orientation and Sustainability

According to Richman et al. [11], employees consider that workplace flexibility has a great effect on them in order to join an organization, be satisfied with the jobs they do, and continue working with the same employers. Employers have become aware of some outcomes, such as being interested, motivated, and retaining their talented employees, having satisfied and numerous engaged employees, along with improving employee effectiveness and success. Moreover, several studies have examined the working skills and characteristics of organizations as a result of the background of organizational retention and engagement, whereas others have examined the primary effect of workplace flexibility. Salvador et al. [12] posited that the flexibility of employees could be clarified and explained in many ways and could be specified from the perspective of workers and from the perspective of organizations. Furthermore, Richman et al. [11] also added that workplace flexibility has a positive connection between the employee and the workplace. Within the general characteristics of employees, which are from small to bigger-sized businesses, flexibility and capability to achieve the requisite balance of work and private life are directly connected with high levels of employee retention and expected engagement. In a flexible workplace, sustainable career satisfaction become important in the long term. Additionally, sustainable career success means employees' personal assessment of their own sustainable career satisfaction. Chang et al. [13] add that there are two kinds of career success. These are objective and subjective career success. Objective points are job titles and rewards gained in yearly, monthly or weekly salaries, which are affected directly and admitted by everyone. On the other hand, subjective career success is personal achievement and personal satisfaction within an achieved career. Even though having a fulfilling career directly increases employee performance, job satisfaction is not the same as long-term career satisfaction. Mahmood et al. [14] expressed that, in today's business, value has become very important during the decision process. That could be more comprehensive where strategies of an organization should be coherent with the performance of employees. Sustainable career satisfaction is long-term satisfaction for a person, whereas job satisfaction is short-term satisfaction within a specific job. However, in pursuing sustainable career satisfaction, stress in the workplace is unavoidable. Wickramasinghe, V. [15] suggested that, usually, job stress is described in the current literature as a feeling of workplace pressure, nervousness, frustration, concern, and suffering at the job. To express this in a different way, variables that are taken into consideration can cause stress. Hence, stressors could be defined as external factors or conditions that have an impact on the individual. Additionally, stress causes deformation or changes in the forces of employees on an individual level. Goal

orientation allows employees to be more focused, which helps to reduce stress. According to Dierendock et al. [16], an achievement goal is concerned with achievement behavior and involves a mixture of beliefs, attributions, and effects that influence behavior intentions; in other words, it is concerned with various approaches to, engagement in, and responses to achievements in various types of activities. As Joo and Park [17] state, having goals at a personal level creates the conditions for employees to have successful goal orientation. In other words, goal orientation is an anticipated variable that motivates by dividing employees' efforts during the learning process. In summary, there are two types of goal orientation. The first is learning, which is based on a task or mastery; the second is social, which is based on performance. Although sustainable career satisfaction is difficult to achieve, the study found that goal orientation helps employees to focus more in a flexible workplace. Finally, Scoones [18] mentioned that the term 'sustainability' must be the most used slogan of the last two eras. In the literature, nothing could be paired with the word 'sustainable', although everything could be matched with it. This will be the first empirical study to combine workplace flexibility, job stress, goal orientation and career satisfaction methodology in the context of the aviation industry. Additionally, there are very few datasets in the literature in which the term 'stress' is used as a positive variable. Moreover, because in the extant literature there is no research that has been carried out regarding the influence of workplace flexibility on sustainable career satisfaction in aviation sector employees, the current research is expected to fill in an important gap in the literature.

### 2.2. Relationship between Workplace Flexibility and Job Stress

Almeida et al. [19] clarified that, recently, the connection between flexibility and stress has become increasingly interesting for researchers. Flexible work strategies have reduced the relationship between stress-related difficulties. Guinot et al. [20] add that stress is not always known as a negative phenomenon. In the current literature, there are two fundamental types of stress: the first is known as eustress, i.e., good stress, and the second is known as distress, i.e., bad stress. Flexibility in the workplace is stressful because of job stress. This means that the relationship between workplace flexibility and job stress is positive. From the eustress point of view, job stress arises especially when employees' talents, dexterity, skills and proficiency can handle the pressure the work stress within the company. In this circumstance, it affects the employees' stress in a good way while handling personal stress in physical and psychological way. Almeida et al. [19] also explain that there is a lack of information in the current literature which shows the link between stress and flexibility. Flexibility might also be known as a defensive aspect against having stress in the daily routine and reacting to stress as a protective factor. Flexibility could be a sustainable solution to employees' distress, to avoid job dissatisfaction. Martínez et al. [21] explain that flexibility of an organization could be protective to achieve a decreasing uncertainty level; however, in the marketplace, it can increase uncertainty levels. Furthermore, Lonnie [22] clarifies that having a set everyday routine work program could have advantages not only for employers, but for employees as well. Similarly, employees might be more productive at the workplace. If they are unable to create their own work schedule, pre-arranging work times may cause differentiation between positive and negative views on work–life balances. Every single employee has their own peak hours, and they have a preference to work at that time for their own efficiency; alternatively, employees could do extra work for their organization at its peak time and trade this extra time devoted to the organization for flexibility in working hours. Employees can do their best to work for a company that provides them with the flexibility they desire.

**Hypothesis 1.** *There is a relationship between four dimensions of workplace flexibility and job stress.*

## 2.3. Relationship between Job Stress and Career Satisfaction

Lounsbury et al. [23] explain how job stress and sustainable career satisfaction could be connected with each other. First of all, especially for mature employees, sustainable careers have become more important at the individual level. At this stage, it is expected that career satisfaction will have a direct relationship with life satisfaction. When this idea is taken into consideration, employees' life satisfaction could influence sustainable career plans, variations, and associated psychological effects of employees. When employees are advancing in their careers, stress could have a direct effect. In addition, Enshassi et al. [24] suggested that while employees are doing their job, there are some ergonomic problems that they face. Among them could be the temperature or the weather. Places with poor lighting and bad site conditions are not the main stressors. The main issues for top management are potentially hazardous work environments. Tausing et al. [25] add that workplace stress is a feature of the job style, such as the presence of a low level of decision making and a large number of demands. Akgunduz, Y. and Eser, S. [26] explained that job stress, which could be explained as a global and severe type of stress, affects employees' energy for performing their job and reduces their capacity. Wickramasinghe, V. [15] also suggests that employees who deal with complex work descriptions, role conflict, work uncertainty, and increased workplace stress could result in having less sustainable career satisfaction. In a similar vein, employees who notify their employers of a lack of flexibility and a lower chance of promotion have lower, albeit more sustainable career satisfaction. Moreover, if the level of role conflict decreases and employees have positive relationships with one another, employees achieve better results in terms of sustainable career satisfaction.

**Hypothesis 2.** *There is a relationship between job stress and career satisfaction.*

## 2.4. Relationship between Workplace Flexibility and Career Satisfaction

According to Vidyarthi et al. [27], employees that have workplace flexibility can achieve high sustainable career satisfaction. A flexible workplace allows employees to make minor adjustments while at work; therefore, employees do not complain about longer work hours because they are already satisfied with their flexible, organized working environment. Thus, employees have better sustainable career satisfaction. In addition, employers' agreement with employees while requesting flexibility, at bottom levels, shows that employees are important in the current market for their employers. It is believed that if employees know how valuable they are to their employers, a satisfactory criterion for employees will become their sustainable career satisfaction. Joo and Lee [28] suggest that, in order to measure employees' sustainable career satisfaction, career success is the most important factor, because it has a direct impact on employees' affirmative psychological conditions while managing their work conditions. There are two kinds of sustainable career success. These are objective and subjective sustainable success. Objective success is all about having a good salary and earning promotions. On the other hand, subjective success pertains to having good job satisfaction and good sustainable career satisfaction. In order to acquire more information about sustainable career satisfaction, in the literature, researchers have explored aspects such as race, characteristic information, and backing from the organization in order to develop better sustainable careers. For example, having strong emotions, determination and fairness were related to better sustainable career satisfaction. Lastly, having good relationships with employers leads to creating better, sustainable career satisfaction.

**Hypothesis 3.** *There is a relationship between four dimensions of workplace flexibility and career satisfaction.*

## 2.5. Relationship between Workplace Flexibility and Career Satisfaction Is Moderated by Goal Orientation

One simple definition of flexibility is that employees are able to choose their daily work and non-work times by themselves. According to current research, the true value of employees' representation is linked to workplace flexibility in rewarding best fit outcomes in the personal, family, and office spheres, as well as in the congregation. Jeffrey et al. [29], in some circumstances, state that choosing the right employee definitely explains workplace flexibility. Exterior organizational factors connected to the real job, at this point, is what businesses require, and the ability to use technology will continuously be the source of apprehension. As a result, some industries will find it easier to adapt to workplace flexibility than others. At some point during the adaptation process, every profession will require long-term career satisfaction. Renee and Bradley [30] clarify that the term "career" is generally used to describe people's work-related knowledge throughout their lives. According to this definition of career, having a successful and sustainable career could be explained as achieving positive job-related results. At this point, a successful career could be defined as rewarding, rank, and promotion in organizations' hierarchy. These jobs, related to positive sustainable career satisfaction, could be affected by goal orientation in order to achieve career success. The success of a career could be explained as psychologically favorable and directly related to the outcomes of work as a cumulative result of employees' work experiences [30,31]. Choi and Nae [32], in compliance with the method of goal achievement, suggest that people could be especially motivated when their selected goals come true. Proportions of goal orientation have been explained in recent studies. Knowing more about goal orientation is the aim for gaining new information or talents, as well as confirming talents and removing negative comments about others' assessments. Therefore, according to goal orientation types, practical sustainable careers become successful in different ways. At first glance, as it is argued in the current literature, people with a quick ability to obtain information about goal orientation and give credence to their skills or talents are unsettled and their interior skills could be increased. Laser, J. [33] states that having extra flexibility and change may affect employees who lack orientation and feel unconfident, because employees' behaviors that become habits no longer appear as successful, and employees begin to improve their current skills in response to change or find fewer opportunities to use skills that become habits. Lee et al. [34] suggest that a goal is a specific action for employees; difficult goals are found to be the antecedent of higher performance than easy goals, and easy goals are not challenging for the employees. For this reason, significant targets should be required in order to achieve their goal. Furthermore, workers who are willing to learn more about goal orientation, are supposedly motivated by their individual growth and progress needs, and as a result, they are looking for opportunities to learn new talents, acquire more information, and master their tasks in order to improve personal enjoyment. Yoo, J. and Jung, Y. [35] clarified that employee empowerment has become very important in recent years regarding goal orientation, and organizations focus on encouraging improved service delivery and performance. Choi and Nae [32] also add that, even if employees fail in their sustainable careers, those that try to manage goal orientation analyze their current situation and try to learn from their mistakes, and acknowledge these situations as a new chance.

**Hypothesis 4.** *The relationship between workplace flexibility and career satisfaction is moderated by goal orientation.*

## 2.6. Relationship between Workplace Flexibility, and Career Satisfaction Is Mediated by Job Stress

Jeffrey et al. [29] explained the main meaning of both occasions for business flexibility and how employees decide to use workplace flexibility for themselves. The main point of the definition is that workplace flexibility is a complex concept that defines workplace performance and the time employees spend completing a given task, as well as the beginning and ending points of rewarded work. Joo and Park [17] claim that there is enough

information in the literature about those affected by these factors. Furthermore, new studies indicate that characteristics based on specific behaviors could have an immediate impact on the workplace. Furthermore, some literature-based research on career success shows that sustainable career success is a long-term behavior, and that the personality of employees might be affected as well. Additionally, employees' perceptions of their present career achievements and their own measurements of their progress towards their career goals define their career satisfaction [36,37]. According to the existing literature, such as Jawahar, I.M. and Liu, Y. [38], an employee's personality is expected to play a significant role in the development of their career satisfaction. As a result, the focus of this research is on long-term employee career satisfaction. When thinking about a career, employees start to have stress in their lives as well. Moreover, Guinot et al. [20] explain that job stress is clarified as the feeling of employees in a negative way. This feeling could be understandable in the workplace. Stress is most commonly observed in situations where the requested work cannot be completed within the employees' abilities. Moreover, if employees' objective and subjective workplaces are mismatched, it can also cause job stress for employees. Chen et al. [39] claim that violent job stress is seen as non-functional and causes employees to decrease their productivity and loyalty. Guinot et al. [20] also add that, from the bad stress point of view, job stress appears in situations where employees' talents, working abilities and skills, workplace pressure, and requested deadlines are not enough sufficient for employers. Stress may also affect employees negatively because they have to control their stress psychologically and physiologically way within the organization.

**Hypothesis 5.** *The relationship between workplace flexibility and career satisfaction is mediated by job stress.*

Figure 1 represents the proposed model of the study. The model was constructed with variables from the literature review based on the hypotheses.

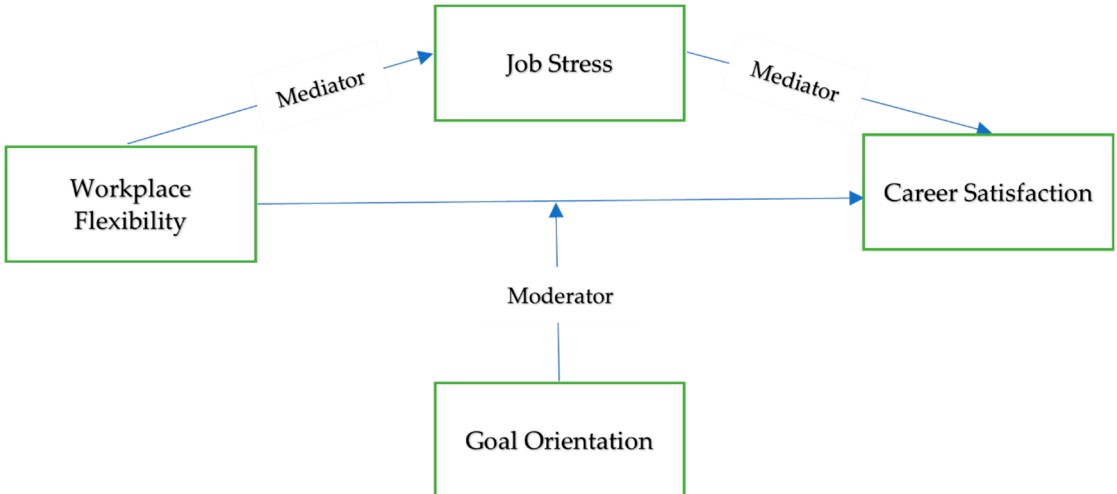

**Figure 1.** Research model of this study.

## 3. Research Methodology

### 3.1. Sample and Procedure

Before distributing the questionnaire to the study sample, the reliability of the questionnaire was tested by using a pilot study. The pilot study proceeded in the same setting as the final study. A total of 30 questionnaires were filled out under the supervision of the researcher face-to-face with respondents, to understand and address missing points or vague communication in the statements. A Cronbach's alpha analysis was used to verify the reliability of the measurement. According to the results of the reliability analysis

of the pilot study, all Cronbach's alpha variables were found to be greater than 0.7. The actual survey session involved the distribution of self-administered questionnaires both online and on paper. The study population sample was all employees currently working in the aviation sector, more specifically, the handling sector, in North Cyprus. In other words, the study adopted a purposive sampling technique to attempt to sample the entire population, which can also be called total population sampling. During data collection, 220 questionnaires were distributed, and 216 usable questionnaires were recovered, which represents a recovery rate of 98.18%. The aviation handling sector in North Cyprus is owned and operated by the government; therefore, regardless of the responsibility level, all workers are considered as employees of the government. On this assumption, the data were collected from all the employees working in the handling sector.

*3.2. Measurements*

The scales used in this study had all been commonly used before in the literature. In order to ensure the reliability of the questionnaire, the statements on the questionnaire were translated and interpreted from their original language (English) to Turkish with the help of English language experts before the questionnaire was distributed. This was performed to ensure the accuracy of the statements and obtain as many reliable answers from respondents as possible.

The questionnaire consisted of two parts. There were demographics questions for the respondents such as age, gender and education, as well as questions regarding their work experience and working hours. Subsequently, the questionnaire included statements about the four variables of workplace flexibility, job stress, career satisfaction and goal orientation. Workplace flexibility has four dimensions including design, time, place and hours. Additionally, questionnaire included five statements regarding workplace flexibility and its influence on employee behavior. Therefore, the first 15 statements were related to workplace flexibility, statements 16–22 were about job stress, 23–27 regarded career satisfaction, and finally, 28–32 concerned goal orientation. A five-point Likert scale method was adopted in the questionnaire, where 1 meant "Strongly Agree", and 5 meant "Strongly disagree".

The statements in the questionnaire were adopted from the references below:

1.  The questions developed regarding flexibility were adapted from Richman et al. [11];
2.  The questions developed regarding job stress were adapted from Goswami et al. [40];
3.  The questions developed regarding career satisfaction were adapted from Chang et al. [13];
4.  The questions developed regarding goal orientation were adapted from Carson et al. [41].

## 4. Results and Figures, Tables and Schemes

*4.1. Demographic Findings*

Table 1 exhibits the demographic information of the respondents.

According to the demographic data, the majority of the respondents were male 72.7%, whereas only 27.3% were female. Additionally, according to the responses, respondents were mainly between 20 and 30 years of age, 22.2% were between 30 and 40 years old, 22.7% were between 40 and 50 years old, and the remaining 16.7% were aged over 50 years. The education level of the respondents was mostly bachelor's degree (47.7%), followed by high school graduates (39.8%), and lastly, master's degree holders comprised 12.5%.

According to the answers of the respondents, as mentioned in Table 2, the majority worked between 40 and 49 h per week (42.6%). A total of 27.3% declared that they worked 39 h or less per week. Among the remaining respondents, 16.7% mentioned that they worked between 50 and 59 h a week, and finally, a minority of the respondents answered that they worked for 60 h or more per week. Additionally, the frequencies showed that a large majority, 65.7%, of the respondents had experience of work in the field for more than

5 years; 29.4% declared that their experience in the field was between 1 and 3 years; and only 10.2% said that they had experience of between 3 and 5 years.

**Table 1.** Demographic information.

| Individual Characteristics | N | Valid Percent |
|:---:|:---:|:---:|
| Age | | |
| 20–30 Years | 83 | 38.4 |
| 30–40 Years | 48 | 22.2 |
| 40–50 Years | 49 | 22.7 |
| Over 50 Years | 36 | 16.7 |
| Total | 216 | 100.0 |
| Gender | | |
| Male | 157 | 72.7 |
| Female | 59 | 27.3 |
| Total | 216 | 100.0 |
| Education Level | | |
| High School | 86 | 39.8 |
| Bachelor's Degree | 103 | 47.7 |
| Master's Degree | 27 | 12.5 |
| Total | 216 | 100.0 |

**Table 2.** Work-related questions.

| Work-Related Questions | N | Valid Percent |
|:---:|:---:|:---:|
| Work Hours per Week | | |
| 39 h or Less | 59 | 27.3 |
| 40–49 h | 92 | 42.6 |
| 50–59 h | 36 | 16.7 |
| 60 h or More | 29 | 13.4 |
| Total | 216 | 100.0 |
| Years of Work Experience | | |
| 1 Year | 8 | 3.7 |
| 1–3 Years | 44 | 20.4 |
| 3–5 Years | 22 | 10.2 |
| More than 5 Years | 142 | 65.7 |
| Total | 216 | 100.0 |

*4.2. Correlations*

According to the correlation analysis in Table 3, workplace flexibility and job stress were found to have a very weak but positive relationship (0.160), whereby the correlation was significant at the 0.05 level. This result was not expected according to the existing literature; however, it may reveal that workplace flexibility may not always reduce job stress, but contribute to the creation of stress in some jobs or circumstances. According to the finding from this correlation analysis, it is possible to conclude that H1 is accepted: "**H1:** *There is a relationship between workplace flexibility and job stress.*" However, it is a weak positive correlation. The next correlation analysis, between job stress and career satisfaction, revealed that the variables had weak and negative relationship between them ($-0.205$).

This finding was expected, and it can be interpreted that an increase in one variable affects the other in a negative way. In other words, job stress is expected to decrease the career satisfaction. According to this result, it is possible to accept H2: "**H2:** *There is a relationship between job stress and career satisfaction*." The relationship between variables is a weak negative relationship. The third correlation analysis was performed between workplace flexibility and career satisfaction. As can be seen from Table 3, weak and positive relationship was found between the two variables (0.295). The finding suggests that an increase in one variable will positively contribute to the other variable. This result supports H3: "**H3:** *There is a relationship between workplace flexibility and career satisfaction*." In addition, and finally, it is possible to see from Table 3 that both workplace flexibility and career satisfaction were positively related with goal orientation (0.225 and 0.365).

**Table 3.** Correlation analysis.

|   |   | **WF** | **JS** | **CS** | **GO** |
|---|---|---|---|---|---|
| WF | Pearson's Correlation | 1 | 0.160 * | 0.295 ** | 0.225 ** |
|   | Sig. (2-tailed) |   | 0.018 | 0.000 | 0.001 |
|   | N | 216 | 216 | 216 | 216 |
| JS | Pearson's Correlation | 0.160 * | 1 | −0.205 ** | 0.219 ** |
|   | Sig. (2-tailed) | 0.018 |   | 0.002 | 0.001 |
|   | N | 216 | 216 | 216 | 216 |
| CS | Pearson's Correlation | 0.295 ** | −0.205 ** | 1 | 0.365 ** |
|   | Sig. (2-tailed) | 0.000 | 0.002 |   | 0.000 |
|   | N | 216 | 216 | 216 | 216 |
| GO | Pearson's Correlation | 0.225 ** | 0.219 ** | 0.365 ** | 1 |
|   | Sig. (2-tailed) | 0.001 | 0.001 | 0.000 |   |
|   | N | 216 | 216 | 216 | 216 |

* Correlation is significant at the 0.05 level (2-tailed); ** correlation is significant at the 0.01 level (2-tailed).

*4.3. Mediation and Analysis*

In the mediation analysis as shown in Table 4, it was assumed that the relationship between workplace flexibility and career satisfaction was mediated by job stress.

**Table 4.** Strength of the effect of workplace flexibility on job stress.

| **Model Summary** |
|---|
| R R-sq MSE F df1 df2 p |
| 0.1605 0.0257 0.6453 5.6560 1.0000 214.0000 0.0183 |
| Model |
| coeff. se t p LLCI ULCI |
| constant 2.2465 0.2487 9.0337 0.0000 1.7563 2.7366 |
| WF 0.2077 0.0873 2.3782 0.0183 0.0356 0.3799 |

According to the results in Table 4, showing the first part of the mediation analysis, the strength of the effect of workplace flexibility on job stress was 0.2077, as shown by the coefficient.

According to the second part of mediation analysis exhibited in Table 5, the effect strength of job stress on career satisfaction was found to be −0.2644 (which can be seen from the coefficient value) with a 0.0001 significance level. Additionally, from Table 5, it is possible to identify that the strength of the direct relationship between workplace flexibility and career satisfaction is 0.4451.

**Table 5.** Effect strength of job stress on career satisfaction.

| Model Summary | | | | | | |
|---|---|---|---|---|---|---|
| R R-sq MSE F df1 df2 p | | | | | | |
| 0.3905 0.1525 0.5879 19.1622 2.0000 213.0000 0.0000 | | | | | | |
| Model | | | | | | |
| coeff. se t p LLCI ULCI | | | | | | |
| constant 1.9193 0.2790 6.8799 0.0000 1.3694 2.4692 | | | | | | |
| WF 0.4451 0.0845 5.2702 0.0000 0.2786 0.6116 | | | | | | |
| JS −0.2644 0.0652 −4.0515 0.0001 −0.3930 −0.1357 | | | | | | |

Finally, in order to fully summarize the hypothesized mediation effect, we examined Table 6, which shows the indirect effect of X on Y, and we can conclude that job stress has a weak negative mediation effect (−0.0549) on the relationship between workplace flexibility and career satisfaction. With this finding, it is possible to accept H5: "**H5:** *The relationship between workplace flexibility and career satisfaction is mediated by job stress.*"

**Table 6.** Analysis of effects of X on Y.

| Indirect Effect(s) of X on Y: | | | |
|---|---|---|---|
| Effect BootSE BootLLCI BootULCI | | | |
| JS −0.0549 0.0321 −0.1267 0.0001 | | | |
| Partially standardized indirect effect(s) of X on Y: | | | |
| Effect BootSE BootLLCI BootULCI | | | |
| JS −0.0662 0.0385 −0.1514 0.0001 | | | |
| Completely standardized indirect effect(s) of X on Y: | | | |
| Effect BootSE BootLLCI BootULCI | | | |
| JS −0.0415 0.0237 −0.0928 0.0001 | | | |

*4.4. Moderation Analysis*

In the moderation analysis, it was assumed that the relationship between workplace flexibility and career satisfaction was moderated by goal orientation.

In the moderation analysis, the regression results showed that our model was significant ($p = 0.0000$) and the R square value showed that our model explained 19.90% of what career satisfaction comprised in this dataset. Additionally, when we examine Table 7, it is possible to state that workplace flexibility is a significant predictor of career satisfaction, as well as goal orientation being explained as a significant predictor of career satisfaction. Lastly, when the interaction (moderation) effect result was evaluated in Table 7, it was found that this effect is significant (0.0294). In other words, goal orientation can be accepted as a moderator in the relationship between workplace flexibility and career satisfaction. Therefore, H4: "**H4:** *The relationship between workplace flexibility and career satisfaction is moderated by goal orientation.*" is accepted.

**Table 7.** Regression analysis for moderation effect.

| Model Summary |
| --- |

| R R-sq MSE F df1 df2 p |
| --- |
| 0.4460 0.1990 0.5582 17.5512 3.0000 212.0000 0.0000 |

| Model |
| --- |

| coeff. se t p LLCI ULCI |
| --- |
| constant −0.2320 0.5939 −0.3907 0.6964 −1.4027 0.9386 |
| WF 0.7314 0.2151 3.4003 0.0008 0.3074 1.1554 |
| GO 0.8988 0.2847 3.1564 0.0018 0.3375 1.4601 |
| Int_1 −0.2187 0.0997 −2.1926 0.0294 −0.4153 −0.0221 |

Additionally, in Figure 2, it is possible to see how goal orientation changes the interaction between workplace flexibility and career satisfaction. According to Figure 2, high levels of goal orientation tend to moderate the interaction between workplace flexibility and career satisfaction. Higher workplace flexibility (WF) produces higher career satisfaction (CS). According to the results, a change in goal orientation will influence the relationship between workplace flexibility and career satisfaction. In other words, it is possible to say that goal orientation enhances the relationship between workplace flexibility and career satisfaction.

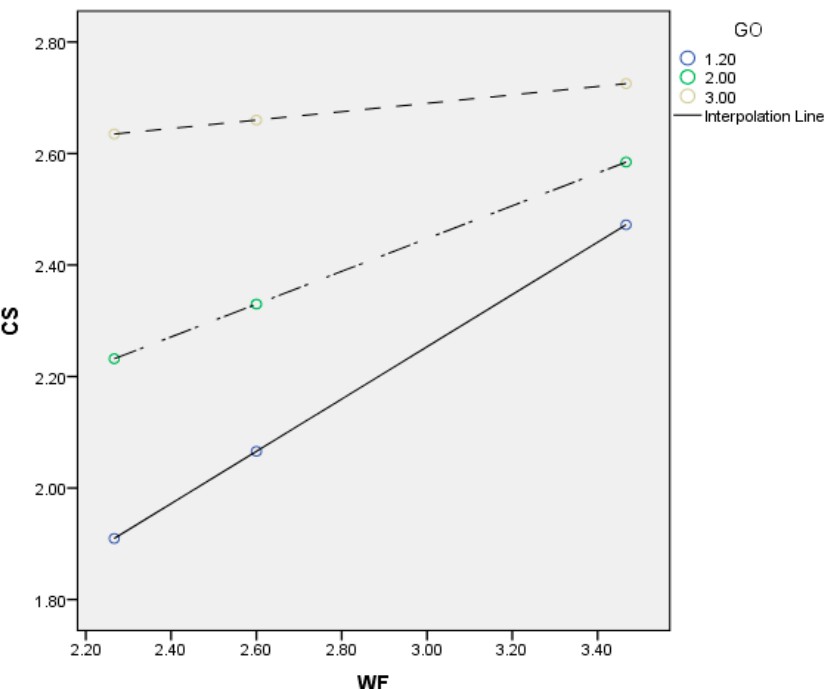

**Figure 2.** Visualizing conditional effect in the moderation analysis.

Table 8 below summarizes the findings and results of the hypothesis testing. As can be seen from the table, all the hypotheses were accepted.

**Table 8.** Hypothesis confirmation table.

| Hypothesis | Accepted/Rejected |
|---|---|
| H1: There is a relationship between workplace flexibility and job stress. | ACCEPTED—weak positive relationship |
| H2: There is a relationship between job stress and career satisfaction. | ACCEPTED—weak negative relationship |
| H3: There is a relationship between workplace flexibility and career satisfaction. | ACCEPTED—weak positive relationship |
| H4: The relationship between workplace flexibility and career satisfaction is moderated by goal orientation. | ACCEPTED—weak positive moderation effect |
| H5: The relationship between workplace flexibility and career satisfaction is mediated by job stress. | ACCEPTED—weak negative mediation effect |

## 5. Discussion and Recommendations

The present study tried to understand the relationship between workplace flexibility and career satisfaction by considering the hypothesized mediation effect of job stress and the moderation effect of goal orientation. Career satisfaction is considered as an important concept in terms of employee happiness and motivation. As a result, ensuring sustainable career satisfaction in the workplace will have a positive effect on employee motivation, which will improve institutional performance. The idea of the present study was to understand how workplace flexibility contributes to career satisfaction and how the other two variables interact with this relationship. For managers, it is crucial to understand these relationships in order to shape management styles and workplace structures accordingly.

The results of the study were derived from tests performed on data collected from employees working in the aviation sector of North Cyprus, specifically in handling. The model of the study hypothesis was supported.

According to the initial results of this study, workplace flexibility and job stress were found to have weak but positive relationship, which was an unexpected result. According to the extant literature, more flexible work environments are expected to decrease jobs stress. In other words, the relationship between workplace flexibility and job stress is expected to have a negative relationship. These findings from the study context could be significant contributions to the existing literature, assuming that employees in the aviation handling sector have different expectations and perceptions about workplace flexibility, as well as other factors that may lead to job stress. When the extant literature is reviewed, the majority of the studies concluded that workplace flexibility decreases stress. However, a few studies had results showing a negative relationship between these two variables. When considering flexibility, we should keep in mind that it is defined in a variety of ways, and it can represent a range of options. For instance, according to Ray and Pana-Cryan's [42] study, which was carried out in 2021, some aspects of flexibility may contribute to stress whereas others can reduce stress: working from home increased the likelihood of job stress by 22%, but changing one's schedule decreased the likelihood of job stress by 20%. On the other hand, another study by Wickramasinghe, V. [15] revealed that work schedule flexibility is negatively associated with job stress. The reason for this negative relationship was explained by Almeida and Davis [19], because those employees with low flexibility are more emotionally and physically reactive to work stressors.

Consequently, job stress and career satisfaction were found to have a weak and negative relationship between them, which was an expected result, considering that consistent job stress can negatively contribute to sustainable career satisfaction. Additionally, in the last correlation analysis, the relationship between workplace flexibility and sustainable career satisfaction was found to be a positive but weak relationship, which reveals that there are other factors contributing to career satisfaction in the long term. In the extant literature, there are many studies supporting this result. For example, Nisar and Rasheed's [43] study in 2020 concluded that stress is an important factor that causes problems with career

satisfaction and job performance in employees. Additionally, Altaf [44] and Coetzee and De Villiers [45] mentioned that perceived sources of job stress are significantly related to employees' career satisfaction.

When the mediation analysis was performed, the results clarify that job stress has a weak negative mediation effect ($-0.0549$) on the relationship between workplace flexibility and career satisfaction. Therefore, because the analysis was found to be significant, it is possible to say that the relationship between workplace flexibility and career satisfaction is mediated by job stress in this case.

Finally, the results from the moderation analysis revealed that workplace flexibility is a significant predictor of sustainable career satisfaction, and goal orientation was found to be a significant predictor of sustainable career satisfaction. The interaction (moderation) effect of goal orientation on the relationship between two variables was significant (0.0294). Additionally, from the results, it is possible to conclude that high levels of goal orientation tend to moderate the interaction between workplace flexibility and career satisfaction. When the extant literature was reviewed, the majority of the studies agreed that workplace flexibility is a predictor of career satisfaction. For instance, according to Prajya et al. [27], flexible work arrangements positively contribute to employees' working conditions, which will lead to higher levels of career satisfaction among employees. Other studies, such as those by Clem et al. [46] and Shauman et al. [47], advocate that workplace flexibility programs are important for career satisfaction and advancement.

## 6. Conclusions

Overall, from the results of this study, it can be concluded that workplace flexibility affects career satisfaction, and this effect is influenced by job stress and goal orientation. As mentioned before, career satisfaction plays a substantial role in the success of employing organizations because it is directly related to how employees feel and behave. Therefore, in this manner, it is important for management to understand what kinds of factors contribute to sustainable career satisfaction.

Rasmussen et al. [48] stated that, in the modern era, global employers are trying to pressure their organizations to have more flexibility in their workplace. However, although workplace flexibility is perceived as a positive factor in decreasing job stress, in some circumstances, such as in the present study, it may cause job stress. Workplace flexibility can have some important drawbacks that can also cause negative impacts an organization. Remote working locations, working from home, or employees working every hour outside of the physical workplace cause problems for employees by increasing their stress and interfering with their family life [49].

On the other hand, goal orientation is always expected to contribute to the sustainability of career satisfaction because it helps employees to focus on both tasks and the end results of the tasks and helps them to motivate themselves. If employees perceive that the end results will contribute to their success and will lead them to better places in their career, they will be more motivated to put in more effort. For instance, Joo and Ready [50] concluded that higher levels of goal orientation contribute to higher levels of career satisfaction for employees.

All these assumptions should be kept in consideration by the management, so that employees will have sustainable career satisfaction, which will positively contribute to organizational performance and success in the long term. With these results in mind, it is recommended that the handling section of the aviation sector in North Cyprus should adopt proactive strategies in order to improve flexible work arrangements by considering the concerns of employees in order to reduce the stress and improve their career satisfaction, which is a contributor to performance (Idowu [51]). Additionally, by considering goal orientation, managers could play important roles in improving career satisfaction of the employees (Joo and Park [17]). Lin, S. and Chang, J. [52] explained that goal-setting is a self-arrangement tactic which has been found to have a positive relationship with performance.

This study was conducted in North Cyprus, a country where there are lots of ongoing embargoes and political conflicts. The country has only one civil aviation airport. Employees from the airport's handling sector were used in this study as the sample. The study sample and location are the major limitations of this research. Researchers tried to collect data from all employees working in the handling sector. However, the results from this study cannot be generalized. Additionally, this study only used the survey method to collect the perceptions of the employees, which may not always be objective data. This can also be considered as a limitation of this study. In order to collect more data from employees through in-depth interviews or observations, future research may use a mixed method approach.

**Author Contributions:** Conceptualization, H.K. and M.Y.; methodology, A.B.; software, H.K.; validation, A.B., H.K. and M.Y.; formal analysis, A.B. and H.K.; investigation, H.K.; resources, M.Y.; data curation, H.K.; writing—original draft preparation, H.K. and M.Y.; writing—review and editing, A.B., H.K. and M.Y.; visualization, H.K. and A.B.; supervision, M.Y. All authors have read and agreed to the published version of the manuscript.

**Funding:** This research received no external funding.

**Institutional Review Board Statement:** Not applicable.

**Informed Consent Statement:** Not applicable.

**Data Availability Statement:** Data are available on request from the corresponding author.

**Conflicts of Interest:** The authors declare no conflict of interest.

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
