# Peer review of "Workplace Flexibility for Sustainable Career Satisfaction: Case of Handling in the Aviation Sector in North Cyprus"

_sustainability, doi:10.3390/su13126878_

Round 1

Reviewer 1 Report

  1. This paper is interesting in area of sustainability in career satisfaction.  This paper will bring benefit to organizations for improving factors that satisfy the career satisfaction of employees and contributes to extent engaged literature.
  2. The paper's argument is built an appropriate base of theory and concept. The design on which the paper is founded, is appropriate to fulfill the author's research hypothesis. But, there are some problems in the questionnaire design content where no find out the occupation, salary, and job position of samples. These demographic information of responder which should be influence the career satisfaction. 
  3. The results need presented and clarify established thoroughly analyzing. The results analysis no find out the hypothesis examination and more discussion for data outcome. Why the results was so conflict and imperfection data.   
  4. This paper needs more citation papers for support the theoretical foundation.

Author Response

Dear Reviewer,

Thank you for your time and efforts, we found your comments very valuable and we tried our best to make all necessary changes and improvements. Please find our answers and comments to your remarks in red. All the changes in the article are marked with Track Changes option.

Thank you,

Authors

Reviewer 2 Report

  1. The title is too long (20 words). Try to shorten it to 12 words, fill it with more meaning. This will significantly increase the interest of readers and further make your research more cited.
  2. Abstracts must be built as follows: Purpose of the article, methods, results, conclusions, and recommendations/future directions. Now the elements of novelty, what is done by the author, are not clear.
  3. I like that the authors describe in great detail the creation of hypothesis structures of this study. But for the correct logical structure of the article, describe the main purpose and objectives of the study. This will simplify the understanding of the article and will allow to track the degree of realization of the hypotheses in the sequence "Result - Conclusions".
  4. On the figures. Figures 1-3 are too simple, the relationships shown logically follow from the text, so the authors should either make them more meaningful, or group them into one. Figure 4 - Give more detailed, more meaningful conclusions after it.
  5. I stress it again that I like the descriptive methodology of the article. But, to my great regret, the results of the study are absent — only tables (1-7) and one figure (Figure 4) from the program are presented. There is no description of the results obtained — the coefficient is so-and-so, and then what? What do these results mean for the author's study? How does this prove the hypotheses put forward? Now the results are inconclusive.
  6. Discussion is completely absent. To justify discussion questions that arose after the study, limitations, prospects, compare them with the results of other authors — this is a discussion.
  7. Conclusions are not related with the study results. Highlight the study purposes in the "Introduction" section, hypotheses — in the "Methods” section, prove them in the "Results" section, and reflect the degree of evidence in the "Conclusions" section. Also, in the "Conclusions" section, highlight the limitations of the study and prospects for further studies. Conclusions must be consistent with the stated purposes of the study. It should also be noted that references are never given in the "Conclusions" section. The "Conclusions" section must be completely redrafted.
  8. Also, the style and language of the scientific article need to be corrected. Pay more attention to the design of the article in accordance with the editorial policy of the journal. For example, References has several design styles and you need to design it in accordance with a single style.

Author Response

(The authors gave the same response as above.)

Reviewer 3 Report

Some suggestions to improve the quality of the article:

  • The Abstract should highlight the findings of the research. Please rewrite.
  • Row 33: any proof? Can it be applied to all employees? Please explain.
  • Row 34: I think you wanted to write employers, not employees.
  • The Introduction section is kind of Literature review. Only opinions of some authors. Please rewrite and explain the effects of flexibility and sustainability in business, in figures, if possible. Please refer to aviation sector, since this is the subject of the study.
  • The purpose of the Literature Review is to find gaps in the area and to fill them with your research. Which are these gaps? Please explain
  • The authors formulated some hypotheses. Are they confirmed by the study?
  • Can the authors provide the questionnaire, or at least explain how the questions were formulated, for which purpose, and how did they analyze the answers?
  • A 98% rate of return of questionnaires is almost unbelievable.
  • Which was the level of responsibility for each respondent? Were they managers, administrative, or others? 
  • The Conclusions Section is very brief, and I don`t understand how it is supported by the results. For example, Row 430 express the behavior of managers. But no managers on the target group, as the table shows.
  • Very poor References.
  • Which are the limitations of the study?

Author Response

Dear Reviewer,

Thank you for your time and efforts, we found your comments very valuable and we tried our best to make all necessary changes and improvements. 

Thank you,

Authors

Reviewer 4 Report

I would like to congratulate the authors for their interest in researching in this field, however, the work presented presents some deficiencies.

a) The proposed title seems to generate some confusion because it does not clearly specify whether the object of the research is workplace flexibility or job stress and moderating Role of Goal Orientation.

The title should have the following characteristics:

-Describe the content of the article in a specific, clear, accurate, brief, and concise manner.

-Enable the reader to identify the topic easily.

-Allow a precise indexing of the material.

I suggest that you slightly revise the title of the article to comply with these premises.

b) Abstract is correct in general terms but does not provide sufficient information on the methods applied to process the information.

Abstract indicates the object of the research, the data acquisition, and a synthesis of the results, but does not describe how the data were actually processed.

Authors should include this information in this section.

c) I found a light description of the survey used in Section 3.1. Although it is not necessary to include the survey itself in the document, the authors should include data that allow the reader to understand the characterization and quantification of the data processed. Authors should expand on this section by providing sufficient information to enable the reader to understand and evaluate the data in this section.

d) I have not seen any preprocessing of the data obtained. They should have filtered out those incomplete surveys, those coming from the same source,... They should explain how they have undertaken this data filtering and, in case they have not done it, implement this pre-filtering. Although chapter 3.2 includes some information on this subject, it is essential that the authors explain in detail the criteria for processing the input data.

e) Discussion section is adequate and extensive, although it does not properly include a comparison with other similar existing studies in order to carry out a real discussion of the results. The authors should expand this section by applying the above considerations.

f) The document does not have a conclusion section. Although this is not a mandatory section, it is advisable, and especially in a research project such as the one presented, it can be a synthesis of the results and reflections of the previous chapter. I encourage the authors to implement this chapter, although I leave the decision to their own discretion.

I hope that these changes will help to improve your article and make it a document of great scientific interest.

Author Response

(The authors gave the same response as above.)

Round 2

Reviewer 2 Report

The authors did a great job of correcting the article, all comments and recommendations were taken into account. I think that the article can be published as is in the journal.

Author Response

Thank you for your comments and all your support during our manuscript review process.

Reviewer 3 Report

The authors provided a much better version of the article.

My only concern is the target group. It`s not structured on levels of responsibility, therefore, in my opinion, the results are not clear.

Author Response

Thank you for your comments and support, due to the fact that, we aim to understand the thoughts and perceptions of employees working in the aviation handling sector, we collected data regardless of the responsibility level of all employees.

We added the highlighted part below in to the methodology part of the article to make it more clear for the reader.

The sample of the study were the all employees currently working in aviation sector, more specifically handling sector, in North Cyprus. In other words, the study adopted a purposive sampling technique which tries to sample the entire population, can also be called total population sampling. During data collection, 220 questionnaires were distributed and 216 usable questionnaires were recovered, which stands for a recovery rate of 98.18%. Due to the fact that, aviation handling sector in North Cyprus is owned and operated by government, regardless of the responsibility level all workers are considered as employees of the government. With this assumption the data was collected from all the employees working in the handling.

Moreover, the results are clarified from the employee perspective, since the perceptions of the employees were taken in to the consideration.

Kind Regards

Reviewer 4 Report

Dear Reviewer,

Thank you for your time and efforts, we found your comments very valuable and we tried our best to make all necessary changes and improvements. Please find our answers and comments to your remarks in red. All the changes in the article are marked with Track Changes option.

Thank you,

I would like to congratulate the authors for their interest in researching in this field, however, the work presented presents some deficiencies.

  1. a) The proposed title seems to generate some confusion because it does not clearly specify whether the object of the research is workplace flexibility or job stress and moderating Role of Goal Orientation.

The title should have the following characteristics:

-Describe the content of the article in a specific, clear, accurate, brief, and concise manner.

-Enable the reader to identify the topic easily.

-Allow a precise indexing of the material.

I suggest that you slightly revise the title of the article to comply with these premises.

Thank you for your suggestion, we shorten the title of article as: Workplace Flexibility for A Sustainable Career Satisfaction; Case of Handling in Aviation Sector in North-Cyprus.

Your correction is as expected. I believe that the new title illustrates the content of the article more adequately.

  1. b) Abstract is correct in general terms but does not provide sufficient information on the methods applied to process the information.

Abstract indicates the object of the research, the data acquisition, and a synthesis of the results, but does not describe how the data were actually processed.

Authors should include this information in this section.

Thank you first of all. Abstract of the paper was improved in line with your suggestions.

Your correction is as expected. The abstract now shows all the necessary information and is much more illustrative.

  1. c) I found a light description of the survey used in Section 3.1. Although it is not necessary to include the survey itself in the document, the authors should include data that allow the reader to understand the characterization and quantification of the data processed. Authors should expand on this section by providing sufficient information to enable the reader to understand and evaluate the data in this section.

Thank you for your comment. The questionnaire was constructed as a result of the literature review. Extant literature was studied and previously utilized set of statements were collected in order to construct the questionnaire. The data was entered into and analyzed by using SPSS programme. With the help of correlation and regression analyses. 

Explanations provided by the authors are enough to clarify this point.

  1. d) I have not seen any preprocessing of the data obtained. They should have filtered out those incomplete surveys, those coming from the same source,... They should explain how they have undertaken this data filtering and, in case they have not done it, implement this pre-filtering. Although chapter 3.2 includes some information on this subject, it is essential that the authors explain in detail the criteria for processing the input data.

Pilot study was done by getting the 14% of the whole population, more specifically 30 respondents. The aim of the pilot study was to find out whether the questionnaire was easy to understand for everyone and also to measure the Cronbach alpha scores of the items to see the reliability level.

In my opinion, this point has not been completely resolved. Autors should indicate how the sample was selected and, in the case of a random process, explain the totality of the potential population to determine that the sample is adequate.

  1. e) Discussion section is adequate and extensive, although it does not properly include a comparison with other similar existing studies in order to carry out a real discussion of the results. The authors should expand this section by applying the above considerations.

Thank you for your comment. The discussion part has been updated with more details. Following the discussion, recommendations and limitations were added. Also, discussion was supported and compared with the results of other studies in extant literature.

Your correction is as expected. Discussion chapter is now complete and shows a real discussion of the results.

  1. f) The document does not have a conclusion section. Although this is not a mandatory section, it is advisable, and especially in a research project such as the one presented, it can be a synthesis of the results and reflections of the previous chapter. I encourage the authors to implement this chapter, although I leave the decision to their own discretion.

I hope that these changes will help to improve your article and make it a document of great scientific interest.

Thank you very much for your comment. The conclusions are included in the discussion section which is improved.

Personally I prefer to separate the discussion and the conclusions but I accept the contributions and explanations of the authors.

Author Response

Thank you for your comments and support, We, as the researchers used convenience sampling technique whereby the researchers tried to reach the whole population which was readily accessible. Due to the fact that, the vice-director in the ministry of civil aviation is a lecturer in the researchers’ university it was convenient to collect data from the majority of population.

Moreover, the discussion is seperated from the conclusion part as you suggest.

Kind Regards